# Influence of Chiral Compounds on the Oxygen Evolution Reaction (OER) in the Water Splitting Process

**DOI:** 10.3390/molecules25173988

**Published:** 2020-09-01

**Authors:** Mirko Gazzotti, Andrea Stefani, Marco Bonechi, Walter Giurlani, Massimo Innocenti, Claudio Fontanesi

**Affiliations:** 1Department of Engineering ‘Enzo Ferrari’, University of Modena and Reggio Emilia, Via Vivarelli 10, 41125 Modena, Italy; gazzotti.mirko@gmail.com (M.G.); 181878@studenti.unimore.it (A.S.); 2Department of Chemistry, University of Firenze, via della Lastruccia 3, 50019 Sesto Fiorentino, Firenze, Italy; marco.bonechi@hotmail.it (M.B.); walter.giurlani@unifi.it (W.G.)

**Keywords:** spin dependent electrochemistry, water splitting, nickel, chirality, OER

## Abstract

Results are presented concerning the influence on the water splitting process of enantiopure tartaric acid present in bulk solution. Stainless steel and electrodeposited nickel are used as working electrode (WE) surface. The latter is obtained by electrodeposition on the two poles of a magnet. The influence and role played by the chiral compound in solution has been assessed by comparing the current values, in cyclic voltammetry (CV) experiments, recorded in the potential range at which oxygen evolution reaction (OER) occurs. In the case of tartaric acid and nickel WE a spin polarization of about 4% is found. The use of the chiral environment (bulk solution) and ferromagnetic chiral Ni electrode allows for observing the OER at a more favorable potential: About 50 mV (i.e., a cathodic, less positive, shift of the potential at which the oxygen evolution is observed).

## 1. Introduction

Nowadays, the energy problem is a crucial challenge to be faced. Hydrogen could be a decisive way out of it, because hydrogen is considered as the ultimate fuel for several reasons: (i) It has the highest specific enthalpy of combustion of any chemical fuel, (ii) it has no carbon or CO_2_ waste, and (iii) water is the combustion’s final product. In the present situation, 80% of the sources of energy production are traditional ones (such as oil, coal, and natural gas). The perspective related to pollution issues and future shortage of oil and coal leads to a search for a different type of fuel. The latter should effectively be coupled with alternative energy sources: Photovoltaic, hydroelectric, wind turbine, geothermal, and tidal power. All these energy sources produce electricity as the final form of energy. On the whole, energy obtained via “renewable energy” sources in 2015 represented the 16.7% of the installed power in the U.S.A. for the 13.8% of the total world-wide electricity generation (including the hydrogen production that reached 10^10^ kg in 2015 in U.S.A) [1,2,3,4]. With a global demand ranging between 70 and 80 million tons of hydrogen per year [5]. In addition, the worldwide percentage of energy produced by renewable energy is higher than in the U.S.A., reaching the 24% of the total [5], with the 58% purchased from hydroelectric source. Within this picture, hydrogen role is growing in importance as it can be used as energy vector and/or fuel mixed with the fossil ones [6,7,8]. Concerning hydrogen, one of the hardest issues to be solved is its efficient storage, which has been the subject of extensive research and multiple solutions are currently viable. From classical liquid storage and high-pressure containers to “Metal hybrids” an “Metal hydride” compounds [9,10]. The hydrogen production via electrochemical dissociation of water is commonly addressed as water splitting (WS) [11,12,13,14,15,16]. Moreover, an appealing positive feature of the hydrogen production obtained via water splitting is its potential use “in-situ” linked to solar energy photovoltaic electric generation in low-cost organic cells [17,18,19]. The efficiency enhancement of water splitting can be achieved by taking advantage of spin filtering of the electron during the reduction process. The control, production, and measurement of spin polarization currents, within an electrochemical system, are studied in the research area of spin dependent electrochemistry (SDE) [20], in which the opportunity to gain further physical insight on the influence and the role of spin in the charge transport is revealed. Chiral organic compound have been discovered performing in a special kind of SDE called Chiral Induced Spin Selectivity (CISS) [21]. When an electron current passes through a chiral molecule, a spin filter effect takes place resulting in an imbalance between the alpha and beta spin of the electrons. This phenomenon has many implications in spintronics as well as in water splitting. The positive effect in WS due to the spin-filter effect has already been demonstrated for electrodes coated with chiral molecules [11,12,22,23,24]. This because “ideal” spin filtering would involve production of OH* radicals only of alpha (or only of beta) spin, hindering the production of H_2_O_2_ (hydrogen peroxide), because two reacting OH* radicals would yield hydrogen peroxide in the triplet state (which is 1 eV higher in energy than the singlet electronic state). The latter is a highly reactive and corrosive compound, and it can be considered a side reaction’s undesirable product of the OH^−^ oxidation. The novelty of this work here consists in exploiting the CISS effect using chiral compounds in bulk solution, rather than adsorbed on the electrode surface (i.e., a bulk rather than an interfacial effect).

Indeed, it is known that the presence in bulk solution of chiral compounds is able to “induce” an enantio-selective or generate local spatial chirality effects: See for instance results obtained in chiral ionic liquids and enantio-recognition effects in achiral ionic liquids, this latter induced by the presence of suitable bulk chiral compounds (even in a low concentration) [25]. In principle, this allows an efficient production of hydrogen through the water splitting, but using a simpler and cheaper system (i) in terms of manufactory costs, being based only on preparing a bulk solution, and (ii) natural chiral compounds, tartaric acid in this specific case, are rather cheap when compared to metallic nanoparticles obtained by both expensive and environmentally-dangerous (for example the cadmium based nanoparticles [11,26]). The water-splitting process is studied as a function of different chiral additives/catalyzers on both a stainless steel and nickel electrodes, the latter deposited directly on a magnetic surface, to evaluate the best candidate for an efficient spin filtering reducing the production H_2_O_2_. In the case of the Ni-on-the-magnet electrode, the ferromagnetic material acts like a spin-injector as a function of the substrate magnetic field; while in the case of the steel electrode, the possible spin filtering is achieved only from the bulk solution chiral environment. The differences between these two different solutions are investigated.

The overall water splitting reaction is, in principle, a simple redox reaction:(1)$$2H_{2}O\to 2H_{2}+O_{2}\;\Delta E=1.23\;V\;\mathrm{vs}.\;\mathrm{RHE}.$$

In fact, the reaction is much more complex, in particular for the water oxidation at the anode. Here, in fact, the oxygen evolution reaction (OER) occurs via a multi-step process, which involves a rather complex reaction mechanism featuring the transfer of 4 electrons [27].

The mechanism depends on a number of chemical as well as physical parameters, and in particular on the solution pH. In the following Equations (2)–(5), the overall half-cell reactions are reported:

Alkaline solution:(2)$$4OH^{-}\to O_{2}+2H_{2}O+4e^{-}\;E_{a}^{0}=0.401\;V\;\mathrm{vs}.\;\mathrm{NHE}$$
(3)$$4H_{2}O+4e^{-}\to 2H_{2}+4OH^{-}\;E_{c}^{0}=-0.828\;V\;\mathrm{vs}.\;\mathrm{NHE}$$

Acid solution:(4)$$2H_{2}O\to O_{2}+4H^{+}+4e^{-}\;\;\;\;\;\;\;\;\;\;\;\;\;\;\;E_{a}^{0}=1.229\;V\;\mathrm{vs}.\;\mathrm{NHE}$$
(5)$$4H^{+}+4e^{-}\to H_{2}\;\;\;\;\;\;\;\;E_{c}^{0}=0.00\;V\;\mathrm{vs}.\;\mathrm{NHE}$$

In principle, the potentials associated to reactions (2) and (4) are the thermodynamic values relevant to the OER, but beyond the thermodynamic potential it is necessary to apply an overpotential to observe the gas evolution. The key to improve the process efficiency is to minimize the overpotential [11,12,13]. The following sequence of elementary reaction steps represents the step-by-step overall reaction that occurs at the anode in an acid solution (where *M* represent the anode metallic conductive substrate) [27,28,29,30].
(6)$$M+H_{2}O_{\left(bulk\right)}\to \left(M\right)OH_{\left(ads\right)}+H_{\left(sol\right)}^{+}+e^{-}\;\;\;\;\;\;\Delta G_{1}$$
(7)$$\left(M\right)OH_{\left(ads\right)}\to \left(M\right)O_{\left(ads\right)}+H_{\left(sol\right)}^{+}+e^{-}\;\;\;\;\;\;\;\;\;\;\;\Delta G_{2}$$
(8)$$\left(M\right)O_{\left(ads\right)}+H_{2}O_{\left(bulk\right)}\to \left(M\right)OOH_{\left(ads\right)}+H_{\left(sol\right)}^{+}+e^{-}\;\;\;\;\;\;\Delta G_{3}$$
(9)$$\left(M\right)OOH_{\left(ads\right)}\to M+O_{2}{}_{\left(gas\right)}+H_{\left(sol\right)}^{+}+e^{-}\;\;\;\;\;\;\Delta G_{4}$$

$\Delta G_{i}$ is the Gibb’s free energy of each elementary steps. As reported in literature [28,29], $\Delta G_{OER}$ is the max of the four $\Delta G_{i}$ values, namely $\Delta G_{3}$. $\Delta G_{i}$ can be calculated with the formula $\Delta G^{0}=-nFE$ if all elementary steps are known and so the $\Delta E^{0}$ of the redox couple involved [31,32,33]. Those values are useful in compiling the Latimer diagram of the reaction, a compact diagram of each step involved in the redox chain, reported in Scheme 1, where numbers in the blue arrows represent the $\Delta E^{0}$ of the reactions from (6) to (9). The $\Delta E^{0}$ of the total reaction is the results of the equation $\frac{\sum_{1}^{4}n_{i}\Delta E_{i}^{0}}{\sum_{1}^{4}n_{i}}$ where $n_{i}$ represent the number of electrons exchanged during each step. 

The energy for the reaction can be provided by different sources, e.g., with the use of light, often coupled with catalytic substrate, like TiO_2_ [34,35,36]. Remarkably, the final product at the anode, i.e., oxygen, is produced in its triplet state, which is the most stable molecular oxygen species. Here spin dependent electrochemistry can play a role. On the whole, the experimental electrochemical outcome allows to discuss the role played on the OER efficiency by the spin filtering effect, in relation to the chirality of compounds present in solution. 

## 2. Results and Discussion

### 2.1. Steel AISI 316L

#### 2.1.1. l-(+)-Tartaric Acid

Figure 1 shows cyclic voltammetry curves, obtained as a function of different concentrations of bulk l-(+)-tartaric acid, on the stainless steel working electrode. In both the forward and backward curves, the current is nearly negligible in the 0.0 to 0.6 V range. In the case of the base electrolyte solution (blue curve, Figure 1), at potential values larger than 0.6 V the current starts to increase almost linearly until reaching a final 2.5 mA value at 0.8 V. Tartaric acid 0.5 mM solution (green curve, Figure 1) shows a lower current, 2.4 mA, at 0.8 V. Tartaric acid 5 mM solution (red curve, Figure 1) shows an initial higher current but at the potential where the OER occurs (around 0.8 V) the current is lower when compared with the previous ones. Tartaric acid 25 mM solution (black curve, Figure 1) shows the lowest current, 1.2 mA, at 0.8 V. The current appears to be correlated to the concentration of the chiral compound with an inverse proportionality. The inset of Figure 1 presents a more detailed CV, in the 0.5 to 0.75 V potential range. The inset shows a small but neat difference between the forward and backward scans. The current for the KOH base electrolyte solution is always higher than the one measured in presence of tartaric acid.

#### 2.1.2. l-(−)-Aspartic Acid

Figure 2 shows CVs in solutions with different concentrations of l-(−)-aspartic acid. Qualitatively, CVs pattern is quite similar to that shown in Figure 1. Quantitatively, all the tartaric acid solutions at different concentration feature a lower current with respect to that of the base electrolyte. Note that, no peaks are evident in the backward scan, indicating that any reaction (OER) occurring during the forward scan (oxidation regime) is irreversible.

CVs recorded by using a stainless steel anode, the l-(+)-tartaric acid and l-(−)-aspartic acid exert a blocking activity on the electrode. The anodic current is found to decrease as the concentration of the acid is increased (compare CVs shown in Figure 1 and Figure 2). Probably this effect is due to simple coulombic attraction between the anode (charged positively) and the relevant acid anion, which can be adsorbed on the surface eventually leading to a decrease in the overall current.

#### 2.1.3. d-(+)-Glucose

Figure 3 shows CVs as function of the concentration of D-(+)-glucose. A systematic variation in the current as a function of the D-(+)-glucose concentration is noted. The maximum efficiency, obtained comparing the current response at fixed potentials, is found for the 10 mM D-(+)-glucose concentration (green curve, Figure 3). The KOH base electrolyte solution is characterized by the lowest current values (red curve, Figure 3).

A complex electrochemical behavior is found when the d-(+)-glucose is present in the solution, Figure 3. The current is always found to be larger than that of the base electrolyte, and in terms of efficiency the current increases as a function of the concentration until 10 mM, then starts to decrease for larger concentrations (compare black line, 100 mM, Figure 3).

### 2.2. Nickel Electrodeposited on Magnet

#### d-(+)-Glucose

Figure 4 shows the cyclic voltammetry of Ni electrodeposited on the north surface of a permanent magnet used as working electrode in two different solutions: d-(+)-glucose 0.1 M in base electrolyte (KOH 0.1 M) and bare base electrolyte. The experimental set-up is the one with the Teflon cell described in Section 3 (3. Experimental). Note that values in Figure 4 and Figure 5 are normalized to point out the differences between the current in the solution with the chiral compound and the one with only the support electrolyte KOH. The value that has been used to normalize the data was the current associated to the oxidation of the Ni surface, otherwise the current peak, in the forward scan, in the base electrolyte curve: An opportune dividing coefficient is adopted in order to make that peak current value equal to 1. In the case of pure electrolyte KOH solution (red curve, Figure 4) the Ni peaks are evident, with the oxidation one happening at 0.55 V in the forward scan while during the backward scan the reduction peak splits in two smaller peaks, one at 0.39 V and the other at 0.46 V. Otherwise, the curve describing the current in the solution containing the chiral compound is completely different, resulting more like the one found with the stainless steel electrode (Figure 3). In particular, the d-(+)-Glucose curve doesn’t have the peaks of the Ni nor in forward nor in backward scan. The absence of peaks in the glucose’s curve means that the redox process became irreversible if the glucose is added to the KOH solution. The anodic peak current in the solution containing the chiral compound results being 50% higher than the one measured with only the support electrolyte.

Figure 5 shows the cyclic voltammetry measured on Ni electrodeposited on the south pole of the permanent magnet used as working electrode; those experiments were carried out with the same solutions of Figure 4, in the same operative conditions. Also in this case d-(+)-Glucose CV curve does not show the characteristic peaks associated to the oxidation and the reduction of the Ni. For what concerns the support electrolyte, the graph shows one peak at 0.55 V in oxidation and one peak and a shoulder in reduction at 0.37 V and 0.45 V respectively, coherent to what has been already seen in Figure 4. Also in Figure 5 the data relevant to the current, ordinate axis, have been normalized using the value of the current associated to the oxidation peak of the support electrolyte, so that peak would result having value equals to 1 on the normalized axis. In comparison with Figure 4, the current of the solution containing the chiral compound is lower in the south pole, resulting having the maximum current in oxidation almost equal to the one measured with the support electrolyte.

Figure 6 shows the comparison between CVs collected with a Ni electrode on top of north and south magnet polarities, recorded in a wider potential range. Remarkably, at 1.2 V it is present a peak for both the curves and the north pole peak current of the oxidation process is larger than the south pole corresponding value.

### 2.3. Step and Sweeps

For a more accurate visual detection of the actual OER starting potential, “step and sweeps” technique measurements were performed. In fact, differently from the previous CVs, these experiments feature incremental values of potential which is maintained constant for a chosen amount of time. These measurements are reported in Figure 7 and Figure 8, showing the anodic current recorded as a function of time during the potential ramp for two different electrodes.

#### 2.3.1. Step and Sweeps on AISI 316 L

Step and sweeps were performed using the AISI 316 L steel sheet as working electrode, with the l-(+)-tartaric acid present in bulk solution. The applied potential program as a function of time features a first potential ramp from 0 to 0.75 V at a 50 mVs^−1^ scan rate, followed by eight +0.01 V potential steps, up to 0.83 V. The potential was maintained constant for 10 s before the next step. Tartaric acid concentration 0.5 and 5 mM in bulk solution was used. A rather large increase in the current was found, which should reflect a more efficient way to produce oxygen. Indeed, visual observation of the anode shows that the oxygen evolution occurs at more negative (smaller) potentials, anticipating the OER between 20 to 30 mV with respect to the pure KOH base electrolyte solution (the relevant table (Appendix A) is present in the Appendix A).

#### 2.3.2. Step and Sweeps Ni-on-magnet

Stepped potential sweeps experiments were recorded also exploiting a ferromagnetic electrode in tight contact with a magnet: The Ni electrodeposited on magnet (Ni-on-magnet) electrode. This to possibly maximize the spin-injection efficiency. Figure 8 shows the anodic current recorded as a function of time: Measurements have been carried out with l-(+)-tartaric acid present in bulk solution, as well as with the pure 0.1 M KOH base electrolyte for reference purposes. The applied potential program as a function of time features a first potential ramp from 0 to 0.70 V at a 50 mVs^−1^ scan rate, followed by +0.01 V potential steps, maintained constant for 15 s. The tartaric acid concentration is 0.5 mM in bulk solution. Results show two different situations depending on the orientation of the substrate magnet. In fact, when the Nickel sheet lays on the south pole of the magnet, the oxidation current results slightly higher for the TA solution than the KOH alone; moreover, a huge current increase can be observed between the chiral solution and the basic one when the North pole of the magnet is used. A significant increase in the current is found, which reflects a more efficient way to produce oxygen. Indeed, for both the North and the South magnet pole visual observation of the anode shows that the oxygen evolution occurs at more negative (smaller) potentials, anticipating the OER between 30 to 50 mV with respect to the pure KOH base electrolyte solution (Appendix A).

In the case of the ferromagnetic Ni surface, deposited on a magnet acting as substrate, is used as the anode. In the case of the l-(+)-tartaric acid, Figure 8, the current recorded in response to potential step and sweeps in the presence of the bulk chiral compound is always larger than that of the base electrolyte. In this peculiar case, the effect of the presence of the chiral compound in bulk solution has been investigated by comparing the peak current values as a function of the magnet orientation (North vs. South), showing significant differences in the values of the current obtained at a fixed potential. Such an analysis is a crucial issue in unravelling the role of the spin. Moreover, to obtain a complete picture, the electrochemical results can be compared for the two enantiomers. Altogether four different situations are to be quantitatively compared: (1) l-(+)-tartaric acid North (2) l-(+)-tartaric acid south (3) d-(−)-tartaric acid north (4) d-(−)-tartaric acid south. Appendix A report, in the most synthetic way, the current results obtained performing CVs in all four cases. Moreover, Appendix A aim to present in a clear and simple way, as much as possible, the catalytic effect observed in the different combination of enantiomers and the magnet pole. In particular, the sign found in column titled “sgn (J_ratio_ North − J_ratio_ South)” (Appendix A) exhibit what is the most efficient combination. Remarkably, the l-(+)-tartaric acid North is more effective than l-(+)-tartaric acid South in a consistent way, while the d-(−)-tartaric acid south combination is found more effective than the d-(−)-tartaric acid North (coherently with the results of the step and sweeps previously presented in Section 2.3.2). In the case of the glucose, only the d-(+)-glucose enantiomer was examined and it is found that the current for the d-(+)-glucose north combination is larger than the d-(+)-glucose south one, Figure 4 and Figure 5. Please note that due to fluctuations in the base electrolyte CVs, in Figure 4 and Figure 5 normalized data are presented. Appendix A report the percentage of spin polarization of currents in the experiments made on the different Ni surface deposited on poles’ magnet. Those values are calculated with the Formula (10).
(10)$$\frac{{\left(\frac{J_{\left(Tart\right)}}{J_{\left(KOH\right)}}\right)}_{\left(North\right)}-{\left(\frac{J_{\left(Tart\right)}}{J_{\left(KOH\right)}}\right)}_{\left(South\right)}}{{\left(\frac{J_{\left(Tart\right)}}{J_{\left(KOH\right)}}\right)}_{\left(North\right)}+{\left(\frac{J_{\left(Tart\right)}}{J_{\left(KOH\right)}}\right)}_{\left(South\right)}}=SP\%$$

## 3. Experimental

### 3.1. Materials

l-(+)-tartaric acid (the natural available enantiomer), l-(−)-aspartic acid, d-(+)-glucose were purchased from Sigma Aldrich and used without further purifications. Only a single enantiomer of each compound is used in the experiments. In fact, if a variation is not applied to the system in order to break its symmetry, measures with opposite sign enantiomers would give the same outcome results. The nickel electrodeposition was carried out using a classical Watt’s Bath (WB), whose composition is: 150 g/L of nickel sulphate, nickel chloride, 37 g/L boric acid H_3_BO_3_, pH = 5.

### 3.2. Methods

Thus, experiments of electrochemical water splitting were carried out using two different anodes: (i) A stainless steel AISI 316 L electrode, and (ii) Ni electrodeposited directly on the North (or South) pole surface of a permanent magnet (Ni-on-the-magnet electrode). Cyclic voltammetry results are presented focusing on the response of the anodic OER. In the case of the AISI 316L steel, results were collected using, l-(+)-tartaric acid, d-(−)-aspartic acid, and d-(+)-glucose. In the case of Ni-on-the-magnet electrode, CV measurements were still performed with bulk concentrations of d-(+)-glucose, l-(+)-tartaric acid, and d-(−)-tartaric acid, and particular focus was also payed to potential sweeps experiments in presence of l-(+)-tartaric acid.

### 3.3. Electrochemical System

Cyclic voltammetry (CV) measurements were performed using both Autolab PGSTAT 128N and CHI660A potentiostats, employing a typical three-electrode electrochemical cell arrangement concerning the measurements involving the stainless-steel working electrode. A different arrangement was adopted in the case of the Ni electrodeposited on the magnet: A Teflon cell, featuring a hole (0.8 cm diameter) in the bottom, was used in a vertical configuration where the Ni-on-the-magnet working electrode was tightened from below, a Teflon ring was used to ensure no solution leakage from the cell. An image is shown in the Appendix A. Thus, Steel AISI 316L or the Ni-on-the-magnet surfaces were used as working electrodes (WE), while a Pt wire and a silver, silver chloride, KCl saturated solution (Ag/AgCl/KCl_sat_) electrodes were the counter (CE) and reference electrodes (RE), respectively. In order to check the quality of the nickel surface obtained via electrodeposition on the magnet, control experiments were carried out using glassy carbon, Pt and evaporated gold as working electrodes [37]. A KOH 0.1 M aqueous solution is used as base electrolyte in all reported electrochemical measurements.

## 4. Conclusions

This work explored the influence of chiral compounds in bulk solution on the hydrogen production process pursued via water electrolysis, often addressed as “water splitting”. To this end two different electrode materials (steel and Ni) and three organic chiral compounds (tartaric acid, aspartic acid, and glucose) were selected. Assessment of the catalytic activity of the different combinations of organic compound and electrode material was performed by measuring CVs under controlled conditions. The KOH 0.1 M aqueous solution base electrolyte was selected as the experimental reference situation to evaluate the catalytic effect on the water splitting process. Chiral compounds have been selected because they are known to enhance the OER efficiency due to spin effects [11,12,13,14,15,16]. The results are encouraging both on steel and on Ni as well. The l-(+)-tartaric acid yields a moderate potential shifting effect (50 mV in the most favorable case), just on the shoulder preceding the current ramp for the OER. On the contrary, the aspartic acid does not seem to exert any prominent effect. In the case of the Ni working electrode the effect of the magnetic field, to select “up” and “down” spin injection, was investigated. Here the most interesting results are obtained. The comparison of the oxidation current peak, essentially due to the oxygen evolution reaction, yields a final consistent picture. Where the l-(+)-tartaric acid is found to enhance the evolution when the north-pole of the magnet is placed in direct contact with the Ni anode surface, the situation is reversed, i.e., larger efficiency for the d-(−)-tartaric acid coupled with the south magnetic field orientation. By and large, it appears that the results obtained using bulk chiral compounds yield a consistent spin filtering effect. On the whole, an average 4% spin polarization value is obtained (average obtained by the SP values of each measurement), a value which is definitively less than the 15% to 20% SP range obtained in the case of well-ordered adsorbed monolayers directly on top of the electrode surface. All in all, the results obtained especially on the Ni as a function of the magnet orientation, relate well with results previously presented in the literature. This gives further impulse to the scientific research in the field of spin effects in the OER process.

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
