# Peer review of "Influence of Chiral Compounds on the Oxygen Evolution Reaction (OER) in the Water Splitting Process"

_molecules, 2020, doi:10.3390/molecules25173988_

Round 1
Reviewer 1 Report
In the manuscript entitled: "Influence of Chiral Compounds on the Oxygen Evolution Reaction (OER) in the Water Splitting Process" the authors have investigated the use of different chiral compounds in the electroyte solution and two different electrodes, stainless steel and Ni electrodeposited on a permanent magnet, for electrochemical OER. The topic is very interesting, timely and worthy to investigate, however, the manuscript cannot be published in the present form since it presents many inconsistences.
For instance, in the introduction section, the hydrogen production as energy vector from renewable sources is widely mentioned, while the role of spin polarization currents in electrochemical systems is mentioned only marginaly. A solid explanation of the role of chiral compounds and the spin filter effect in electrochecmial systems, with significant examples, would be very convenient.
In the results section, a working electrodes made of stainless steel and Ni electrodeposited on a magnet have been carried out. Despite the use of magnetic and non-magnetic working electrodes is necessary to study the spin filter effect, similar materials must be employed in order to achieve fair comparison between the electrodes. For that reason, the stainless steel electrode should be coated with Ni by electrodeposition with the aim to obtain identical surface than in the case of the Ni coated magnet.
On the other hand, L-tartaric acid, L-aspartic acid and D-glucose have been use with the stainless steel electrode in order to study the effect of chirality. However, they all are different molecules, and therefore they exhibit different behaviour. It would be more convenient to use L and D-taqrtaric acid, and/or L and D-aspartic acid and L and D- glucose to obtain fear comparison, and obtain information only from the chiral effect. Besides, the Ni electrodeposited on magnet electrode has been only used with D-glucose, sutidng the effect of north and south poles. In order to obtain general conclusions, these experiments should be carried out with L and D- glucose, as well as with the other compounds. Moreover, the stepped sweeps experiments have been carried out only with L-tartaric acid with the two electrodes, and the current trend was the opposite fo the CVs in previous experiments. How do the authors explain that the current in the stepped sweeps experiments is higher with 5 mM of L-tartaric acid than the bare electrolyte, while the CV shows the opposite. Why there is almost no difference in current between 5 and 0.5 mM of L-tartaric acid, while in the CV there was?
Finally, the discussion section was found very descriptive, but critical results discussion is missing. Which is the role of chirality in this study? which is the influence of magnetic field? an scheme of the role of the L vs. D chirality would be helpful as well as an small table with the OER onset potentials obtained from the different experiments to illustrate the conclusions.
In overall, this investigation is very interesting but further work is needed before publication in this journal.
Reviewer 2 Report
This paper talks about the effect of the chiral compound such as L(+) tartaric acid on the oxygen evaluation reaction.
General Comment:
Your study should emphasize the novelty of the research over the existing work. There is ample scope to improve the paper content, and I am recommending major revisions for the introduction and result and discussion part.
Please rewrite the introduction by mentioning the novelty of the work. Merge the Result and discussion parts. Also, provide a few experimental results that can prove that chiral compounds can improve the oxygen evaluation reaction.
Before publication, the paper needs to be read by a native English speaker.
Specific comment:
Introduction section
- The introduction needs to be improved. The story should be systematic: it started mentioning SDE without any background; then suddenly express about energy issues; after that talked about CISS. The transition of these topics is not very smooth. In my opinion, you need to rewrite the introduction and mention properly about the novelty.
- Rewrite the following sentence:
“Spin dependent electrochemistry (SDE) is a quite recently proposed method branch of electrochemistry [1]”.
- For the following, please use more recent statistical data instead of 2015 data:
“On the whole, energy obtained via “renewable energy” sources in 2015 represented the 16.7% of the installed power in the U.S.A. for the 13.8% of the total world-wide electricity generation [8] (including the hydrogen production that reached 10 10 kg in 2015 in U.S.A.)”.
- Please elaborate on the ‘positive boost’ for the following
“The positive boost due to the spin-filter effect has already been demonstrated for electrodes coated with chiral molecules [2,3,18–20]”.
- What is the novelty of this work?
- You mentioned that your system is ‘cheaper’; where is the evidence?
Experimental section
- Need subheadings: material, method, and instrument;
- Provide the picture of the experimental setup;
- Elaborate about the experimental details for water splitting
Results section
- In the results section, you talked about water splitting background, overpotential, and finally the spin-dependent electrochemistry. This part should be in the introduction section to explain the novelty of your work.
- The following sentences should be in the method section:
“Thus, experiments of electrochemical water splitting were carried out using two different anodes: i) a stainless steel AISI 316 L electrode ii) Ni electrodeposited directly on the North (or South) pole surface of a permanent magnet (Ni-on-the-magnet electrode). Cyclic voltammetry results are presented focusing on the response of the anodic OER. In the case of the AISI 316L steel, results were collected using, L-(+)-tartaric acid, D-( )-aspartic acid and D-(+)-glucose. In the case of Ni-on-the-magnet electrode, CV measurements were still performed with bulk concentrations of D-(+)-glucose, L-(+)-tartaric acid and D-( )-tartaric acid, and particular focus was also payed to potential sweeps experiments in presence of L-(+)-tartaric acid”.
- Explain what SDE is? And why it will be useful for water splitting?
- I saw the error message for citation on several occasions “Error! Reference source not found”. Please correct all such citations.
- Instead of a separate discussion section, please discuss the results in the result section for each plot.
Round 2
Reviewer 1 Report
In the new version of the manuscript the authors have improved significantly the quality of the manuscript, especially in the introduction and conclusion sections, which are now much more clear and make the manuscript easy to follow and the main objectives are clearly distinguished. In addition, the experimental data have been properly structured. For that reason, I recommend for publication in the present form.
Reviewer 2 Report
Comments:
- As I commented before on the citation error, “Error! Reference source not found”. It is still there. Also, the same kind of error is seen in Figure captions. Please address those issues.
- Figures 4, 5, and 6: the legend boxes are hiding Y-axis.
- Equation numbers (1), (2), (3), and (4) are repeated. Check Line number 84-86, pg 2 &3.
- Line 21, pg 1: replace ‘spin dependent’ with ‘spin-dependent’; do the same throughout the manuscript
- replace ‘water splitting’ with ‘water-splitting’; do the same throughout the manuscript
- Please use articles in the document where needed. For example, Line 34, pg1: ‘2015 in USA’ should be ‘2015 in the USA’. Please check for other cases where you missed articles.